# A Comprehensive Evaluation of Minimally Invasive Achilles Tendon Reconstruction with Hamstring Graft Indicates Satisfactory Long-Term Outcomes

**DOI:** 10.3390/medicina58101417

**Published:** 2022-10-09

**Authors:** Bartosz Kiedrowski, Paweł Bąkowski, Paweł Cisowski, Łukasz Stołowski, Jakub Kaszyński, Magdalena Małecka, Tomasz Piontek

**Affiliations:** 1Department of Orthopedic Surgery, Rehasport Clinic, 60-201 Poznań, Poland; 2Sport Championship School on the Handball Federation in Poland, 09-402 Płock, Poland; 3Department of Spine Disorders and Pediatric Orthopedics, University of Medical Sciences, 61-701 Poznań, Poland

**Keywords:** Achilles tendon tears, minimally invasive Achilles tendon reconstruction, a comprehensive evaluation of Achilles tendon

## Abstract

*Background and objectives:* The Achilles tendon, the largest tendon in the body, is vulnerable to injury because of its limited blood supply and the combination of forces to which it is subjected. Given the relevance of the Achilles tendon in the proper function of the foot and ankle, the primary goal of the present study was to use a holistic approach for a comprehensive evaluation of Achilles tendon reconstruction results on multiple levels. *Materials and Methods:* The study was designed in the following way: 30 patients with partial or total Achilles tendon tears were subjected to the minimally invasive Achilles tendon reconstruction. Patients were then subjected to the clinical, functional and isokinetic tests 12 and 24 months after the treatment. The clinical evaluation included calf circumference measurements and subjective patient-reported tests: ATRS, EQ-5D-5L and VAS scales. The functional evaluation was based on three tests: the weight-bearing lunge test, the heel rise test and single leg hop. Isometric and isokinetic evaluation was performed using a Biodex 3 dynamometer. *Results:* The calf circumference of the operated limbs was significantly lower than the non-operated limb 12 months after the surgical procedure, however this improved at the second evaluation. All subjective outcomes improved significantly 24 months after the surgery. Significantly better results in the function of the operated limbs were also obtained 24 months after the surgery. However, most of the muscle strength parameters of the operated limbs were already comparable to non-operated ones 12 months after the surgery and were comparable between two evaluation times. *Conclusions*: The overall results of this extensive evaluation are highly satisfactory and patients returned to their normal physical activity. From a medical point of view, it is assumed that the healing process is completed 12 months after the surgery, however, importantly, our results indicate that we should consider the healing process and the rehabilitation process separately.

## 1. Introduction

The Achilles tendon is the largest and strongest tendon in the body. A partial or a total rupture of this tendon can occur with a sudden strain or a stretch. The incidence of Achilles tendon ruptures is 18–40 per 100,000 persons per year and occurs more frequently in the fourth decade of life [1,2]. The repair of the ruptured Achilles tendon is critical for the proper function of the lower limb. For years, the Achilles tendon has been repaired through a large open incision. This technique required full tendon exposure and the repair was executed in the damaged area of the tendon. Recently, percutaneous, minimally invasive repair techniques are receiving more interest, since the surgery is completed through small incisions and the stitches are placed in the strong, healthy area of the tendon. Moreover, the tendon exposure is no longer necessary and this decreases the scar formation and wound-healing issues and allows more rapid recovery, earlier physical therapy and a quicker return to activity. However, in the case of Achilles tendon re-ruptures, degeneration or chronic tears, when the tendon tissue is damaged, a reconstruction technique using a graft is needed.

Despite the surgical method used, the outcomes of the Achilles tendon repair have to be certainly evaluated. Multiple clinical and functional examination procedures have been designed for that purpose. This includes patient-reported subjective tests (Hannover Achilles Score, Leppilahti Score, Rupp Achilles Tendon Score, Achilles Tendon Total Rupture Score, American Orthopedic Foot and Ankle Society Score and more [3]) and objective clinical evaluation by a physiotherapist or an orthopedist. Moreover, the characteristics of the skeletal muscles under dynamic conditions can be objectively measured using specialized devices, isokinetic dynamometers. However, it has been recently recognized that there is no consensus on the best or most appropriate way to measure the strength after Achilles tendon reconstruction, irrespective of the treatment [4]. No unified protocol exists for establishing strength after Achilles tendon reconstruction.

In our recent research, we showed that a minimally invasive, endoscopic Achilles tendon reconstruction using semitendinosus and gracilis tendons with Endobutton stabilization allowed for good functional recovery [5,6]. As a logical consequence, the primary goal of the present study was to use a holistic approach for a comprehensive evaluation of Achilles tendon reconstruction results on multiple levels: clinical, functional and isokinetic. Given the relevance of the Achilles tendon in proper function of the foot and ankle, we reasoned that such exhaustive evaluation of the recovery of motor function requires not only a clinical or subjective assessment, but also a quantitative analysis of muscle strength and endurance. To our knowledge, this is the first report aimed at revealing the results of both subjective and objective tests performed in isolated and functional conditions. Moreover, we have separately evaluated the results obtained by patients with a partial and a total Achilles tendon tears.

## 2. Materials and Methods

### 2.1. Design of the Study

The aim of the study was to prospectively evaluate the results of surgical treatment of Achilles tendon performed endoscopically with the use of autologous semi-tendon (ST) or gracilis (GR) tendon grafts. Surgical treatment results were assessed on the basis of patients’ evaluations, which took place 12 and 24 months postoperatively. During the evaluation visit, the patients underwent medical and physiotherapeutic evaluation. The medical examination was aimed, in particular, at assessing the current ailments, range of motion, superficial sensation and the circumference of the lower extremity. The physiotherapeutic examination consisted of a functional, isometric and isokinetic assessment. The study was conducted in accordance with the Declaration of Helsinki, and approved by the Bioethics Committee of Karol Marcinkowski Medical University in Poznań, by resolution no. 743/20. Informed consent was obtained from all subjects involved in the study.

### 2.2. Patients Characteristic and Treatment

A total of 30 patients with Achilles tendon tears were included in the study. All 30 patients were treated surgically with a minimally invasive Achilles tendon reconstruction technique using a hamstring autografts, as previously described [5]. The diagnosis of Achilles tendon tears were based on magnetic resonance imaging. The surgical procedures were performed by two trained orthopedic surgeons (PB and TP) between 2016 and 2019 at Rehasport Clinic in Poznań. All patients followed the proprietary rehabilitation protocol developed at our Clinic [7].

The inclusion criteria were as follows: (i) age> 18 years old; (ii) age <65 years old; (iii) written consent to participate in the study; (iv) Achilles tendon injury (neglected rupture of the Achilles tendon with retraction of the stumps over 3 cm, chronic partial damage to the Achilles tendon involving more than 50% of the tendon fibers, previous failure of conservative and surgical treatment) treated with a minimally invasive Achilles tendon reconstruction technique using a hamstring auto-grafts. The exclusion criteria involved: (i) contraindications for MRI performance (ferromagnetic implants, dentures, implants); (ii) active infection process; (iii) metabolic diseases.

The whole group of 30 patients was examined 12 months after the procedure. Additionally, at this time-point we distinguished and evaluated separately two subgroups: (i) seventeen patients, who suffered from a partial tendon rupture, and (iii) thirteen patients, who had a total tendon injury. In all, 9 patients did not participate in the second evaluation, 24 months after the procedure, therefore only 21 patients from the starting group were additionally evaluated 24 months after the procedure. Due to the lower subgroup’s sizes (13 patients with partial and 8 with total tendon rupture), the evaluation was only performed for the whole group. The demographic data of the patients enrolled in the study are presented in Table 1.

### 2.3. Clinical and Subjective Evaluations

The clinical examination was performed by a trained physiotherapist (BK) and included calf circumference measurements. The measurements were performed as follows: the patient was placed on a chair and held their bare foot down, holding the leg folded to 90°. The circumference of the calf was measured at its widest point, laying the tape on the skin without tightening. Additionally, the subjective outcomes of the surgery reported by patients were recorded, as previously described. These tests were based on the pain and the satisfaction levels and included the visual analogue scale (VAS scale), the Achilles tendon Total Rupture Score (ATRS) [8] and the quality of life questionnaire (EQ-5D-5L) [9].

### 2.4. Functional Evaluations

The functional evaluation was performed by a trained physiotherapist (BK) and was based on three tests: the weight-bearing lunge test, the heel rise test and single leg hop. All tests were performed for each limb separately, following the existing protocols [10]. The weight-bearing lunge test was performed to measure the extent of dorsiflexion of the ankle joint and foot complex [11]. The heel rise test was performed to measure the endurance of the Achilles [12]. A single leg hop test was used for measuring the distance of the horizontal jump [13].

### 2.5. Isometric and Isokinetic Evaluation

Isometric and isokinetic evaluation was performed using a Biodex 3 dynamometer as previously descried [14]. Both the injured and non-injured legs were tested by trained physiotherapists to assess the isometric, concentric and eccentric muscle-strength parameters. An average isokinetic knee extension and flexion peak torques were measured at concentric and eccentric velocity of 30°/s. The total contractional work, the average power (the average time rate of work) of the muscles, as well as the maximum peak torque (angle peak torque) were appointed at 90°/s velocity. The explosive power of the muscles (time to peak torque) was evaluated at three concentric and eccentric velocities: 30°/s, 60°/s and 90°/s. The lowest velocity allowed for measuring the muscle strength, the median–the speed and the higher–the muscle endurance. Therefore, the testing started with the 30°/s angular velocity and then increased to higher velocities. An average of three repetitions was performed for each angular velocity for both the affected and intact limbs. Before testing, practice trials were given to the patients to familiarize them with the machine.

### 2.6. Statistical Analysis

The statistical analysis of the results was carried out in the R environment. For statistical testing, the paired Wilcoxon signed rank test was used. For all measurements, statistical significance was considered for *p* < 0.05. Statistical characteristics of individual measurements were carried out through the analysis of the distribution of values. The distribution was non-normal and the data are presented as median ± standard deviation. Spearman correlation tests between parameters were performed.

## 3. Results

### 3.1. Clinical and Subjective Evaluations

The results of the calf circumference examination are presented in Table 2. We have observed a statistically significant difference (*p* = 0.0479) in the circumference of the operated vs. non-operated limb 12 months after the surgical procedure. This difference was only observed in the total Achilles rupture subgroup (*p* = 0.0161). There was no statistically significant difference in circumference between the operated and non-operated limbs after 24 months. The injured leg showed slightly decreased maximum calf circumference than the opposite leg in almost all cases, but the difference was not significant.

The subjective outcomes reported by patients at 12 and 24 months after the surgical treatment are gathered in Table 3. The results obtained in the subjective tests did not differ statistically between the partial and the total Achilles tear subgroups. The ATRS score improved significantly 24 months after the surgery (*p* = 0.009), with four patients reaching the maximum score of 100 points. The median value of the EQ-5D-5L scale improved statistically (*p* = 0.0087), and the EQ-5D-5L health comfort test also increased significantly, from 75.0 to 90.0 (*p* = 0.0008). The median reported pain and satisfaction level according to the VAS scale was 1.0 (±1.5) and 9.0 (±1.4) 12 months after the procedure, respectively, and 0.0 (±0.9) and 10.0 (±0.4) at the second evaluation (*p* = 0.007 and *p* = 0.0143, respectively). The maximum level of VAS satisfaction score (10 points) was recorded by 10 participants after 12 months postoperatively and 13 at 24 months follow-up.

### 3.2. Functional Evaluation

We have observed statistically significant differences in performing all functional tests with an injured extremity between the two follow-ups (Table 4). The results of the functional evaluation of non-operated limbs did not change significantly between 12 months and 24 months after the surgery. Generally, significantly better results in the function of the operated limbs were obtained 24 months after the surgery. However, a significant difference was still observed in the functional tests at both follow-ups, when we compared the operated vs. non-operated limb (with favoring of the non-operated one), both in the partial and the total tear subgroups. Patients were able to reach the larger distance from the wall to the tip of the big toe at 24 months follow-up (when compared to 12 months, *p* = 0.0118) during the weight-bearing lunge test with the operated limb. A similar situation was observed in the number of repetitions of the heel rise in the heel rise test (*p* = 0.0001). The largest difference in the number of repetitions performed with the operated limb at 12 months after the surgery was 14 (19 for the uninjured limb and 5 for the injured limb in two cases). At 24 months after the surgery, we observed clearly smaller differences: the same number of heel rise repetitions was observed for non-operated and operated limb in one patient and a difference of one heel rise in six cases. The differences in the distance of the injured (*p* = 0.0037) and non-injured (*p* = 0.0253) leg hop were statistically significant when we compared 12 months vs. 24 months postoperatively, with better results obtained at the second evaluation.

### 3.3. Isometric Evaluation

Lower limb strength is dependent on the body weight. Therefore, the limb strength was measured as an average peak torque (the turning effect of a given force on an object) in two motions: away from and towards the dynamometer, and expressed in relation to the patient body weight, in order to compare the results between individuals (Table 5). A statistically significant improvement was observed for the strength of the operated limb, measured in the “away” motion between the first and the second evaluation (*p* = 0.0364). The average peak torque/body weight in the “toward” motion was comparable for operated and non-operated limbs in partial tear and total tear subgroups.

### 3.4. Isokinetic Evaluation

Most of the muscle strength parameters were comparable between two evaluation times (Table 6, *p* value n.s.). A statistically significant improvement was observed both during the knee flexion and extension in operated limb, between the first and the second evaluation in the peak torque/body weight parameter (*p* = 0.0048 and 0.0065, respectively). Similar results were also noticed for the non-operated limb. The average power of the muscles in the operated extremities significantly improved through the course of evaluations (*p* = 0.0342). However, the average muscle power during the knee extension did not improve and the differences between the operated and non-operated limb were observed during both evaluations. The same situation was observed for the total work measurements (both evaluations), and time to peak torque during knee extension at 30°/s velocity at the 24 months follow-up (*p* = 0.0263) and at 60°/s velocity at the 12 months follow-up (*p* = 0.0260).

Importantly, there were no statistically significant differences in any of the isokinetic tests performed on the operated limbs, between the partial and the total Achilles tear subgroups. A significant deterioration was observed for the operated limbs in the total tear subgroup in multiple isokinetic parameters (Table 7).

### 3.5. Correlations between the Subjective and Objective Tests

Importantly, there were two strong correlations between the subjective and the objective tests. The VAS score (pain assessment) results highly correlated with the heel rise endurance test (r = −0.5457, *p* = 0.0016) and the results of the EQ-5D-5L score strongly correlated with the average power of the muscles during knee flexion (r = −0.5306, *p* = 0.0023).

## 4. Discussion

The most important finding of our work on 30 patients is that the minimally invasive Achilles tendon reconstruction using semitendinosus and gracilis tendons with Endobutton stabilization might enable a satisfactory restoration of strength and function of the operated area within 24 months, as well as the overall satisfaction of patients. From a medical point of view, it is assumed that the healing process is completed 12 months after the surgery. However, importantly, our results indicate that the healing process might take longer than 12 months.

The results presented in our study are in a good agreement with already published ones. The medical literature describes many methods of Achilles tendon reconstruction using autologous grafts of different origin [15,16,17,18,19,20]. The use of the semitendinosus and gracilis tendon-derived grafts possess the advantage that they do not directly alter foot function [21]. Moreover, it has been proven that better repair effects through external implementation of growth factor and/or stem cell therapy (during the tendon healing) can be achieved [22]. Several authors presented restoration of full function of the operated Achilles tendon and the overall patient satisfaction when using semitendinosus tendon grafts [18,23,24,25] or gracilis tendon grafts [26,27]. However, to our knowledge, this is the first study which presents a detailed, versatile clinical, functional and isokinetic assessment of patients after Achilles tendon reconstruction. Multiple assessments, which described the outcomes of the Achilles reconstruction were based on two most popular subjective, patient-reported scales: Achilles tendon Total Rupture Score (ATRS) and American Orthopedic Foot and Ankle Society Score (AOFAS), combined with isokinetic tests or functional tests [15,16,17,18,19,22,23,24,25,26,27]. However, in these studies, a clear contradiction was observed between the subjective and the objective outcomes, which, in our opinion, might be the result of an insufficient number of the Achilles tendon parameters tested. For example, in 2014, Zhao et al. observed that some of the isokinetic parameters were significantly reduced in the operated limb [20]. Nevertheless, the patients did not report any deficits in function and were satisfied with the return to the physical fitness, despite clear deficits in the isokinetic and clinical assessment. In another study, two isolated isokinetic parameters were evaluated (peak torque of the plantar and dorsal flexor) and the authors observed two contradictory results for two knee positions [16]. All the above results indicate the need for a multifaceted approach to the Achilles tendon evaluations.

In the present study, the overall satisfactory outcomes are the result of multiple parameters: (i) equalization of the calf circumference between operated and non-operated limb; (ii) a statistically significant improvement in the subjective scores (ATRS, VAS and EQ-5D-L); (iii) better performance in functional tests (weight-bearing lunge test, heel rise test, single leg hop); (iv) a statistically significant improvement in the isometric strength of the operated limb, measured in the “away” motion; and (v) a statistically significant improvement in multiple lower limb muscles’ strength and endurance isokinetic parameters.

However, the results of almost all parameters of the operated limbs did not reach the values similar to non-operated ones at the first evaluation time point. The calf circumference values, which reflect calf muscle function, are usually smaller in the operated limbs due to the limb immobilization after the surgery. At 12 months after the Achilles tendon surgery we still observed that the operated limbs were significantly narrower than non-operated ones, but only when the tendon was ruptured totally. However, patients with the total tendon rupture reported the same satisfaction levels as the patients with the partial Achilles tendon tear, at both follow-ups. Importantly, patient-reported outcomes indicated their better overall health and activity 24 months post-operatively, when compared to 12 months follow-up. At the second evaluation, patients reported: (i) less limitations and difficulties related to the injured Achilles tendon (ATRS score); (ii) better mobility and self-care, easier performing usual activities (EQ-5D-5L); and (iii) less pain and discomfort (EQ-5D-5L and VAS), when compared to the 12 months follow-up.

Generally, significantly better results in the function of the operated limbs were obtained 24 months after the surgery. This was measured with three parameters: the extent of dorsiflexion of the ankle joint and foot complex (the weight bearing lunge test), the Achilles endurance test (the heel rise test) as well as the distance of the horizontal jump (the single leg hop test). The strength of the muscles in the limbs was tested with the isokinetic parameters: an average peak torque and a time to peak torque at 30°/s velocity. A statistically significant improvement was observed for the strength of the operated limb between the first and the second evaluation when measuring the average peak torque, in one motion (toward the dynamometer device) during both muscle flexion and extension. Moreover, the muscle strength measured with a time to peak torque at 30°/s velocity parameter was comparable for the operated and non-operated limb already at the first follow-up, 12 months after the surgery. A similar situation was observed when we tested the total contractional work of the muscles, but the average power of the muscles in the operated limbs reached the non-operated limb levels only after 24 months. The muscle endurance (measured with a time to peak torque at 90°/s velocity) was functionally restored already 12 months after the surgery.

All of the above results lead us to the conclusion that the Achilles tendon function was properly restored 24 months after the surgery. Moreover, a detailed examination of the partial and the total tear subgroups led us to the conclusion that the minimally invasive Achilles tendon reconstruction using hamstring grafts enables satisfactory restoration of strength and function, despite the severity of the damage.

## 5. Conclusions

The conclusion of our research of 30 patients is the confirmation of the high levels of satisfaction and function with the minimally invasive Achilles tendon reconstruction using hamstring grafts, the absence of the functional limitations and the return to pre-injury activity, regardless of the severity of the initial Achilles damage.

## Figures and Tables

**Table 1 medicina-58-01417-t001:** Demographic data of the patients enrolled in the study.

	MIN	MED	MAX	SD
**(i)** **Partial tear subgroup (*n* = 17)**				
Body weight (kg)	63.0	80.0	125.0	12.8
Height (cm)	167.0	176.0	194.0	8.5
Age (years)	21.0	57.0	67.0	12.7
**(ii)** **Total tear subgroup (*n* = 13)**				
Body weight (kg)	65.0	85.0	125.0	16.3
Height (cm)	170.0	178.0	190.0	5.9
Age (years)	32.0	53.0	74.0	13.6

A minimal value (MIN), a median value (MED), a maximal value (MAX) and standard deviation (SD) are presented.

**Table 2 medicina-58-01417-t002:** Clinical evaluation of calf circumference.

Calf Circumference [cm]	12 Months (*n* = 30)	24 Months (*n* = 21)	*p* Value
non-operated limb	37.0 ± 2.7 (33.0–43.0)	37.5 ± 2.8 (33.0–43.0)	n.s.
operated limb	38.5 ± 3.2 (30.0–41.5)	38.0 ± 3.0 (32.0–44.0)	n.s.
***p* value**	0.0479	n.s.	
	**partial tear (*n* = 17)**	**total tear (*n* = 13)**	***p* value**
non-operated limb	37.0 ± 2.7 (34.0–42.5)	38.5 ± 3.9 (33.0–46.5)	n.s.
operated limb	36.5 ± 3.1 (32.5–42.5)	38.0 ± 3.8 (30.0–46.0)	n.s.
***p* value**	n.s.	0.0161	

Values are presented as median ± standard deviation. The minimum and maximum values are given in brackets. *p* value is presented. n.s.—*p* value not significant.

**Table 3 medicina-58-01417-t003:** The subjective outcomes of the surgical procedure.

Test	12 Months (*n* = 30)	24 Months (*n* = 21)	*p* Value
ATRS	79.5 ± 12.7 (58.0–99.0)	90.0 ± 9.5 (72.0–100.0)	0.0009
EQ-5D-5L	6.0 ± 2.6 (5.0–14.0)	6.0 ± 2.0 (5.0–13.0)	0.0097
EQ-5D-5L Health today	75.0 ± 15.9 (30.0–100.0)	90.0 ± 14.1 (40.0–95.0)	0.0008
VAS pain	1.0 ±1.5 (0.0–5.0)	0.0 ± 0.9 (0.0–3.0)	0.0007
VAS satisfaction	9.0 ± 1.4 (5.0–10.0)	10.0 ± 0.7 (8.0–10.0)	0.0143
	**partial tear (*n* = 17)**	**total tear (*n* = 13)**	***p* value**
ATRS	84.0 ± 14.4 (50.0–99.0)	86.5 ± 24.8 (7.0–96.0)	n.s
EQ-5D-5L	6.0 ± 2.2 (5.0–14.0)	6.0 ± 1.2 (5.0–9.0)	n.s
EQ-5D-5L Health today	80.0 ± 16.4 (30.0–100.0)	80.0 ± 11.9 (65.0–100.0)	n.s
VAS pain	1.0 ±1.7(0.0–5.0)	0.0 ± 1.0 (0.0–3.0)	n.s.
VAS satisfaction	9.0 ± 1.1 (7.0–10.0)	10.0 ± 1.1 (7.0–10.0)	n.s.

Values are presented as median ± standard deviation. The minimum and maximum values are given in brackets. *p* value is presented. n.s.—*p* value not significant.

**Table 4 medicina-58-01417-t004:** The functional outcomes of the surgical procedure.

	12 Months (*n* = 30)	24 Months (*n* = 21)	*p* Value
**weight bearing lunge test**			
non-operated limb	13.0 ± 2.9 (5.0–18.0)	12.0 ± 2.0 (5.0–19.0)	n.s.
operated limb	10.0 ± 2.8 (2.0–14.0)	10.0 ± 2.5 (5.0–15.0)	0.0118
***p* value**	0.0002	0.0030	
**heel rise test**			
non-operated limb	16.0 ± 6.2 (2.0–22.0)	16.0 ± 5.4 (3.0–26.0)	n.s.
operated limb	9.0 ± 6.0 (0.0–21.0)	13.0 ± 5.4 (3.0–22.0)	0.0001
***p* value**	0.0000	0.0000	
**single leg hop**			
non-operated limb	100.0 ± 27.5 (40.0–140.0)	110.0 ± 34.0 (45.0–190.0)	0.0235
operated limb	79.0 ± 29.7 (40.0–145.0)	105.0 ± 35.7 (40.0–180.0)	0.0037
***p* value**	0.0010	0.0063	
	**partial tear (*n =* 17)**	**total tear (*n =* 13)**	***p* value**
**weight bearing lunge test**			
non-operated limb	12.0 ± 2.7 (5.0–16.0)	12.0 ± 3.2 (3.0–16.0)	n.s.
operated limb	10.0 ± 2.4 (4.0–13.0)	7.0 ± 3.5 (2.0–14.0)	n.s.
***p* value**	0.0064	0.0063	
**heel rise test**			
non-operated limb	16.0 ± 6.1 (2.0–22.0)	17.0 ± 5.8 (4.0–22.0)	n.s.
operated limb	10.0 ± 6.5 (0.0–21.0)	9.0 ± 5.4 (0.0–15.0)	n.s.
***p* value**	0.0000	0.0008	
**single leg hop**			
non-operated limb	97.5 ± 36.0 (40.0–175.0)	100.0 ± 35.8 (45.0–150.0)	n.s.
operated limb	82.5 ± 36.4 (40.0–165.0)	178.0 ± 33.6 (45.0–140.0)	n.s.
***p* value**	0.0066	0.0013	

Values are presented as median ± standard deviation. The minimum and maximum values are given in brackets. *p* value is presented. n.s.—*p* value not significant.

**Table 5 medicina-58-01417-t005:** The isometric outcomes of the surgical procedure.

	12 Months (*n =* 30)	24 Months (*n =* 21)	*p* Value
**average peak torque/body weight away (%)**			
non-operated limb	1.2 ± 0.3 (0.4–1.6)	1.1 ± 0.3 (0.6–1.9)	n.s.
operated limb	0.8 ± 0.4 (0.3–1.7)	1.0 ± 0.3 (0.4–1.6)	0.0364
***p* value**	0.0001	0.0008	
**average peak torque/body weight toward (%)**			
non-operated limb	0.6 ± 0.1 (0.4–0.7)	0.6 ± 0.2 (0.3–1.1)	n.s.
operated limb	0.6 ± 0.2 (0.1–0.8)	0.6 ± 0.2 (0.4–1.0)	n.s.
***p* value**	n.s.	n.s.	
	**partial tear (*n =* 17)**	**total tear (*n =* 13)**	***p* value**
**average peak torque/body weight away (%)**			
non-operated limb	1.1 ± 0.3 (0.4–1.5)	1.2 ± 0.3 (0.7–1.6)	0.0174
operated limb	0.8 ± 0.4 (0.3–1.7)	0.9 ± 0.4 (0.1–1.6)	n.s.
***p* value**	0.0327	0.0041	
**average peak torque/body weight toward (%)**			
non-operated limb	0.6 ± 0.1 (0.3–0.7)	0.6 ± 0.2 (0.4–0.7)	n.s.
operated limb	0.6 ± 0.1 (0.2–0.8)	0.6 ± 0.1 (0.4–0.7)	n.s.
***p* value**	n.s.	n.s.	

Values are presented as median ± standard deviation. The minimum and maximum values are given in brackets. *p* value is presented. n.s.—*p* value not significant.

**Table 6 medicina-58-01417-t006:** The comparison of the isokinetic outcomes of the surgical procedure at two evaluation times.

			12 Months (*n =* 30)	24 Months (*n =* 21)	*p* Value
**peak torque/body weight (%)**					
**flexion**	**30°/s**	non-operated limb	1.8 ± 0.6 (0.0–2.6)	1.9 ± 0.5 (1.2–3.0)	0.0089
		operated limb	1.5 ± 0.6 (0.5–2.5)	1.8 ± 0.5 (1.1–2.8)	0.0048
		*p* value	n.s.	n.s.	
**extension**	**30°/s**	non-operated limb	1.7 ± 0.7 (0.0–2.7)	2.0 ± 0.5 (1.3–3.1)	0.0139
		operated limb	1.5 ± 0.6 (0.2–2.5)	1.8 ± 0.5 (1.2–3.0)	0.0065
		*p* value	n.s.	0.0475	
**total work (J)**					
**flexion**	**90°/s**	non-operated limb	472 ± 200 (150–798)	533 ± 221 (183 0 956)	n.s.
		operated limb	325 ± 218 (164–941)	462 ± 161 (140–699)	n.s.
		*p* value	0.0246	n.s.	
**extension**	**90°/s**	non-operated limb	528 ± 251 (247–1053)	585 ± 271 (213–1065)	n.s.
		operated limb	425 ± 235 (94–839)	488 ± 222 (228–1193)	n.s.
		*p* value	0.0341	0.0088	
**average power (W)**					
**flexion**	**90°/s**	non-operated limb	42.8 ± 17.6 (12.8–72.1)	43.9 ± 16.5 (18.0–80.7)	n.s.
		operated limb	29.0 ± 17.3 (17.1–69.5)	42.3 ± 12.0 (13.4–58.6)	0.0342
		*p* value	0.0318	n.s.	
**extension**	**90°/s**	non-operated limb	42.6 ± 22.8 (23.4–101.0)	46.0 ± 19.8 (22.7–86.3)	n.s.
		operated limb	41.1 ± 20.5 (11.7–80.9)	42.7 ± 17.1 (22.2–95.8)	n.s.
		*p* value	0.0300	0.0101	
**angle peak torque (deg)**					
**flexion**	**90°/s**	non-operated limb	−10 ± 7 (−30–2)	−13 ± 4 (−23–−4)	n.s.
		operated limb	−12 ± 9 (−35–12)	−12 ± 5 (−23–−6)	n.s.
		*p* value	n.s.	n.s.	
**extension**	**90°/s**	non-operated limb	−10 ± 8 (−30–10)	−13 ± 4 (−10–−4)	n.s.
		operated limb	−12 ± 10 (−35–21)	−12 ± 4 (−23–−6)	n.s.
		*p* value	n.s.	n.s.	
**time to peak torque (msec)**					
**flexion**	**30°/s**	non-operated limb	10 ± 132 (10–420)	10 ± 55 (10–250)	n.s.
		operated limb	10 ± 98 (10–440)	10 ± 91 (10–410)	n.s.
		*p* value	n.s.	n.s.	
	**60°/s**	non-operated limb	10 ± 76 (10–320)	10 ± 43 (10–190)	n.s.
		operated limb	10 ± 68 (10–310)	10 ± 43 (10–200)	n.s.
		*p* value	n.s.	n.s.	
	**90°/s**	non-operated limb	10 ± 55 (10–250)	10 ± 37 (10–170)	n.s.
		operated limb	10 ± 41 (10–190)	10 ± 41 (10–190)	n.s.
		*p* value	n.s.	n.s.	
**extension**	**30°/s**	non-operated limb	1290 ± 392 (150–1960)	1320 ± 324 (700–1830)	n.s.
		operated limb	1250 ± 355 (230–1900)	1170 ± 305 (760–1750)	n.s.
		*p* value	n.s.	0.0263	
	**60°/s**	non-operated limb	830 ± 222 (280–1230)	850 ± 158 (600–1170)	n.s.
		operated limb	840 ± 217 (130–1200)	880 ± 162 (590–1180)	n.s.
		*p* value	0.0260	n.s.	
	**90°/s**	non-operated limb	660 ± 162 (160–950)	680 ± 186 (570–1410)	n.s.
		operated limb	660 ± 159 (130–930)	690 ± 268 (520–1720)	n.s.
		*p* value	n.s.	n.s.	

Values are presented as median ± standard deviation. The minimum and maximum values are given in brackets. *p* value is presented. n.s.—*p* value not significant.

**Table 7 medicina-58-01417-t007:** The comparison of the isokinetic outcomes of the surgical procedure between two Achilles tear subgroups.

			Partial Tear (*n =* 17)	Total Tear (*n =* 13)	*p* Value
**peak torque/body weight (%)**					
**flexion**	**30°/s**	non-operated limb	1.6 ± 0.4 (1.2–2.6)	1.8 ± 0.4 (1.2–2.4)	n.s.
		operated limb	1.4 ± 0.5 (0.9–2.4)	1.7 ± 0.6 (0.7–3.2)	n.s.
		*p* value	0.0303	n.s.	
**extension**	**30°/s**	non-operated limb	1.7 ± 0.4 (1.2–2.7)	2.0 ± 0.4 (1.2–2.5)	n.s.
		operated limb	1.5 ± 0.5 (1.0–2.4)	1.7 ± 0.6 (1.0–3.3)	n.s.
		*p* value	0.0340	n.s.	
**total work (J)**					
**flexion**	**90°/s**	non-operated limb	395 ± 195 (150–802)	485 ± 158 (315–798)	n.s.
		operated limb	329 ± 216 (92–941)	335 ± 180 (110–704)	n.s.
		*p* value	n.s.	n.s.	
**extension**	**90°/s**	non-operated limb	533 ± 191 (120–721)	688 ± 265 (329–1023)	0.0047
		operated limb	421 ± 182 (154–822)	451 ± 202 (192–776)	n.s.
		*p* value	n.s.	n.s.	
**average power (W)**					
**flexion**	**90°/s**	non-operated limb	33.1 ± 14.5 (12.8–63.1)	39.4 ± 14.9 (27.4–72.1)	0.0457
		operated limb	30.0 ± 15.8 (7.2–69.5)	32.6 ± 15.7 (10.9–61.7)	n.s.
		*p* value	n.s.	0.0000	
**extension**	**90°/s**	non-operated limb	43.6 ± 16.2 (8.9–60.7)	62.5 ± 20.3 (30.9–90.5)	0.0008
		operated limb	38.7 ± 15.8 (11.7–71.0)	44.9 ± 17.2 (18.5–67.2)	n.s.
		*p* value	n.s.	0.0000	
**angle peak torque (deg)**					
**flexion**	**90°/s**	non-operated limb	−14 ± 7 (−30–−6)	−13 ± 5 (−20–−5)	n.s.
		operated limb	−12 ± 9 (−35–−5)	−13 ± 3 (−17–−8)	n.s.
		*p* value	n.s.	n.s.	
**extension**	**90°/s**	non-operated limb	−14 ± 7 (−30–−5)	−13 ± 5 (−19–−5)	n.s.
		operated limb	−12 ± 8 (−35–−4)	−12 ± 3 (−17–−8)	n.s.
		*p* value	n.s.	n.s.	
**time to peak torque (msec)**					
**flexion**	**30°/s**	non-operated limb	10 ± 32 (10–140)	10 ± 104 (10–310)	n.s.
		operated limb	10 ± 8 (10–40)	10 ± 0 (10–10)	n.s.
		*p* value	n.s.	n.s.	
	**60°/s**	non-operated limb	10 ± 31 (10–130)	10 ± 43 (10–150)	n.s.
		operated limb	10 ± 11 (10–50)	10 ± 7 (10–30)	n.s.
		*p* value	n.s.	n.s.	
	**90°/s**	non-operated limb	10 ± 55 (10–230)	10 ± 3 (10–20)	n.s.
		operated limb	10 ± 16 (10–60)	10 ± 5 (10–30)	n.s.
		*p* value	n.s.	n.s.	
**extension**	**30°/s**	non-operated limb	1430 ± 331 (940–1960)	1370 ± 205 (1050–1760)	n.s.
		operated limb	1335 ± 307 (850–1900)	1190 ±183 (1000–1530)	n.s.
		*p* value	n.s.	0.0089	
	**60°/s**	non-operated limb	885 ± 194 (570–1230)	860 ± 193 (670–1040)	n.s.
		operated limb	850 ± 180 (560–1200)	800 ± 82 (690–950)	n.s.
		*p* value	0.0200	n.s.	
	**90°/s**	non-operated limb	715 ± 130 (480–950)	660 ± 75 (540–790)	n.s.
		operated limb	640 ± 120 (480–930)	660 ± 91 (510–840)	n.s.
		*p* value	n.s.	n.s.	

Values are presented as median ± standard deviation. The minimum and maximum values are given in brackets. *p* value is presented. n.s.—*p* value not significant.

## Data Availability

The datasets used and analyzed during the current study are available from the corresponding author on reasonable request.

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
