# Peer review of "A Comprehensive Evaluation of Minimally Invasive Achilles Tendon Reconstruction with Hamstring Graft Indicates Satisfactory Long-Term Outcomes"

_medicina, 2022, doi:10.3390/medicina58101417_

Round 1

Reviewer 1 Report

-          The design of the study must be shown in the abstract.

-          “Vas” score should be replaced by VAS score within the abstract.

-          The design of the study should be shown at the beginning of the Method section, as well as the fact that this research followed proper guidelines and ethics.

-          Participants in the study are very poorly described. Inclusion-exclusion criteria must be clearly expressed.

-          How was the diagnosis of the patient carried out?

-          How were the assessments performed and who did it?

-          Were the patients previously treated in rehabilitation and physiotherapy departments?

-          Was the surgery offered as the last treatment option?

-          Pearson´s correlations were performed, however, the distribution of the data was non-normal, therefore Spearman´s test should be used instead.

-          Is the sample size properly calculated? 11 subjects per group is lower than usual. I see the authors used an 80% confidence level. Why not report power?

-          How was the distribution of the data performed?

-          Subheading 3.5 has a typo in “correlations”.

-          Correlations are poorly expressed. The p-value of each correlation should be shown.

-          To assure that “minimally invasive Achilles tendon reconstruction using semitendinosus and gracilis tendons with Endobutton stabilization enables satisfactory restoration of strength and function of the operated area within 24 months, as well as the overall satisfaction of patients, is daring. Only 30 subjects participated in this study and only the report from your results should be expressed.

-          Readers can easily get confused when reading “From a medical point of view, it is assumed that the healing process is completed 12 months after the surgery, however, importantly, our results indicate that we should consider the healing process and the rehabilitation process separately”. What do you specifically mean? Is not improving the healing process one of the main goals in rehabilitation or the rehabilitation itself?

-          Discussion section lacks references and therefore must be strengthened importantly.

-          The conclusion must be changed since only 30 subjects participated in the study.

Reviewer 2 Report

Dear Authors,

thank you for the submitted manuscript, I appreciated it for completeness and clarity.

In order to improve it further, I recommend that you to mention and comment on the paper "Indino C, et al. Biologics in the Treatment of Achilles Tendon Pathologies. Foot Ankle Clin. 2019 Sep;24(3):471-493. doi: 10.1016/j.fcl.2019.04.009. Epub 2019 May 21. PMID: 31370998."

Apart from this point, the manuscript in my opinion deserves publication in this valuable journal.

Reviewer 3 Report

I read this article titled “A comprehensive evaluation of minimally invasive Achilles tendon reconstruction with hamstring graft indicates satisfactory long-term outcomes”. The authors present a data about objective and subjective outcomes after minimally invasive Achilles tendon reconstruction with hamstring graft in a relatively small cohort of patients. 

A few issues must be addressed before considering the paper suitable for publication:

1) Please add the data about a calf circumference measurement method

2) In statistics: It would be better to add number of participants (n) in each subgroup in all tables.

3) In statistics: I have a doubt athe bout significance in table 2, part 2 (total tear NonOperated - 38.5 ± 3.9 (33.0 – 46.5), Operated -38.0 ± 3.8 (30.0 – 46.0) where you calculated p level about 0,0161... Can you provide the original data you reached as a supplementary file of the article or for the purposes of this review? 

4) In my opinion - for the better flow of the article, it would be better to connect all graphs together in one for each category separately (Subjective outcomes, Functional outcomes, Clinical evaluation, Isometric outcomes, Isokinetic outcomes). Also, it is not necessary to provide a graphs with nonsignificant results - the data are already in the previous table.

Round 2

Reviewer 1 Report

The authors have addressed the comments mostly.

Now the manuscript has increased the quality.

Congratulations.

Author Response

We are grateful for this comment. 

Reviewer 3 Report

I agree with the author's replies to my comments.

Author Response

We are grateful for this comment.